# Effect of Heat Treatment on Microstructure and Mechanical Properties of a Selective Laser Melting Processed Ni-Based Superalloy GTD222

**DOI:** 10.3390/ma14133668

**Published:** 2021-06-30

**Authors:** Tian Xia, Rui Wang, Zhongnan Bi, Guoliang Zhu, Qingbiao Tan, Rui Wang, Ji Zhang

**Affiliations:** 1High-Temperature Materials Department, Central Iron & Steel Research Institute, Beijing 100081, China; xiatian@cisri-gaona.com.cn (T.X.); wangrui1029@sjtu.edu.cn (R.W.); wangrui@cisri-gaona.com.cn (R.W.); zhangji@cisri-gaona.com.cn (J.Z.); 2Shanghai Key Laboratory of Advanced High-Temperature Materials and Precision Forming, School of Materials Science and Engineering, Shanghai Jiao Tong University, Shanghai 200240, China; qbtan1981@sjtu.edu.cn

**Keywords:** selective laser melting, heat treatment, nickel matrix superalloy, carbide, mechanical properties

## Abstract

Additive manufacturing (AM) of nickel-based superalloys is of high interest for application in complex hot end parts. However, it has been widely suggested that the microstructure-properties of the additive manufacturing processed superalloys are not yet fully clear. In this study, the GTD222, an important superalloy for high-temperature hot-end part, were prepared using selective laser melting and then subjected to heat treatment. The microstructure evolution of the GTD222 was investigated and the mechanical properties of heat treated GTD222 were tested. The results have shown that the grain size of the heat treated GTD222 was close to its as-built counterparts. Meanwhile, a large amount of γ’ and nano-scaled carbides were precipitated in the heat treated GTD222. The microstructure characteristics implied that the higher strength of the heat treated GTD222 can be attributed to the γ’ and nano-scaled carbides. This study provides essential microstructure and mechanical properties information for optimizing the heat treatment process of the AM processed GTD222.

## 1. Introduction

Many complex metal parts of the aero-engine and gas turbine need to work in harsh environments, such as working blades and guide blades [1,2]. However, the traditional processing methods, such as casting and forging, were hard to achieve near-net shaping of complex metal components due to the reduced material utilization and long processing progress [3]. The additive manufacturing (AM) techniques have become the research hotspot owing to their ability to efficiently fabricate parts in low volumes, minimize subtractive processing steps, and access complex geometries [4,5,6]. During several AM technologies, selective laser melting (SLM) has attracted more attention than other techniques [5,7,8]. This advanced process builds the parts in a layer-by-layer manner and it is particularly suitable for fabricating parts with complex fine features [5]. SLM provides high design flexibility for net-shaped parts with high precision, for instance, turbine blades with complicated internal cooling channels [8].

Superalloys were widely used to prepare hot-end parts due to their outstanding high-temperature mechanical properties and excellent resistance of oxidation [3,9,10]. One of the Ni-based superalloys with the best comprehensive properties over the past three decades has been the GTD222, which was developed as the replacement of the GTD111 [11,12,13]. GTD222 was a γ’ strengthened nickel-based superalloy and it can remain its high microstructure stability and excellent mechanical properties at a temperature as high as 760 °C [12,14]. Meanwhile, the processability and weldability of the GTD222 were outstanding due to the low volume of Al and Ti. The cooling rate of metals during the SLM process range from 10^3^–10^6^ k/s, which was much faster than casting [15,16]. During the repeatedly rapid heating and cooling, the plastic deformation, which may affect the accuracy of components, is hard to avoid owing to the thermal stress [7,17,18,19]. Thus, it was indispensable to modify the microstructure and mechanical properties of the SLM processed GTD222 through heat treatment. Ramakrishnan et al. [20] found that the mechanical properties of the AM processed Haynes 282 superalloy were superior to its casting counterparts. The increased volume of carbide is regarded as the most influential factors. In our previous work, the effect of direct aging on the microstructure and mechanical properties of the AM processed GTD222 was carried out. The results show that the strengths of the GTD222 increased with decreasing ageing temperature, while the elongation declined simultaneously. The precipitation of γ’ was affected by the aging but no carbides were precipitated after direct aging [21]. The effect of the carbides on mechanical properties was not clear yet.

In this work, the solution + aging heat treatment was performed to regulate the microstructure of the SLM processed GTD222. The microstructure and mechanical properties of the as-built and heat-treated (HTed) GTD222 were investigated. The evolution of the microstructure and mechanical properties were discussed. This work can aid in optimizing the heat treatment process of the SLM processed GTD222.

## 2. Experimental

The SLM processed superalloy GTD222 investigated in this study was fabricated using EOS M 290 additive manufacturing system (EOS GmbH Electro Optical Systems, Munich, Germany). The GTD222 powders with a particle diameter distribution of 15–53 μm were Ar gas atomized (GAed) in the Shanghai Materials Research Institute (Shanghai, China). The chemical compositions and the size distribution of the GAed GTD222 superalloy powers can be founded in our previous work [13]. Before SLM process, the powders were dried at 150 °C for 4 h under vacuum. The size of the as-built GTD222 samples was 13 mm*13 mm*75 mm. The SLM process parameters were set as below: Laser power 250 W, laser scanning speed 960 mm/s, hatching space 80 μm, and layer thickness 20 μm. The GTD222 powders were spread on a stainless-steel substrate (Bao wu steel, Shanghai, China). The stainless-steel substrate was preheated to 150 °C. The whole manufacturing of the GTD222 was performed under Ar condition. The porosity of the GTD222 samples were investigated using X ray microscope (XRM, Xradia 520 Versa, Carl Zeiss, Jena, Germany), more details could be founded in our previous research [13].

The heat treatment of the SLM processed GTD222 was carried out in a horizontal cylindrical furnace (SKL-1700, Shenyang, China) under Ar atmosphere. The heat treatment process for the SLM processed GTD222 was 1170 °C, 4 h/air cooling, 800 °C, 8 h/ air cooling. The heat treatment parameters used in the current study were the optimized heat treatment parameters suitable for the casting GTD222.

The microstructure of the SLM processed GTD222 samples before and after tensile testing were investigated using the scanning electron microscope (SEM, JEOL JSM-7800F, Tokyo, Japan), equipped with an electron back-scattered diffraction (EBSD) detector. The SEM samples were etched in a Cr_2_O_3_ corrosive liquid (Sinopharm Group, Beijing, China) at 7 V for 3 s. The EBSD samples were polished using vibration polishing machine (Vibrotech, Shanghai, China) with diamond polishing fluid for 2 h. The substructure of the samples was also investigated using the transmission electron microscope (TEM, Talos F200X G2, Thermo Fisher Scientific, Waltham, MA, USA). The TEM samples were electrochemically polished with a twin-jet polisher (Shanghai Jiao Tong University, Shanghai, China), with 8 vol% HClO_4_ + 92 vol% CH_3_O_4_ at −30 °C. The tensile testing was performed using the Zwick Amsler 100 HFT 5100 testing machine (Zwick Roell Group, Ulm, Germany) and the testing temperature was 760 °C. The tensile rate was 0.0001/s and at least five samples were tested. The tensile specimens were cut from the as-built GTD222 samples and the dimension of the tensile specimen was given in Figure 1.

## 3. Results and Discussion

The inverse profile figure (IPF) of the as-built and HTed GTD222 samples were shown in Figure 2. Figure 2b shows that the HTed GTD222 was consisted of the columnar grains, which was similar to the as-built GTD222, as shown in Figure 2a. In addition, the statical results indicate the average grains of the HTed GTD222 was 37 μm, while the average grain diameter of the as-built samples was 36.3 μm. According to Figure 2, the difference in grain between HTed and as-built GTD222 was subtle, suggesting the HT has little effect on grain morphology and size. The fact that columnar grains of the HTed samples were similar to the as-built samples implies that the static recrystallization was not take place during the heat treatment. Similar results have been observed in previous works reported by G. P. Dinda [20,22]. According to the results of Dinda, the columnar grain in the as-built IN625 samples cannot disappear until 1000 °C [23]. The recrystallized grains of as-built IN625 were fully formed at 1200 °C, which was much higher than the recrystallization temperature of IN625 processed by traditional technology ranges from 930 to 1040 °C. Hu et al. [3] believe that the recrystallization temperature of the as-built superalloy was higher than that of the traditional deformed superalloy. In addition, the static recrystallization temperature of AM processed superalloys was closed to that of the traditional deformed superalloy has been observed. Although equiaxed grains were helpful to uniform mechanical properties, eliminating anisotropy of alloy and make its properties more predictable [24,25]. In this study, static recrystallization of the SLM processed GTD222 was not observed after heat treatment and no equiaxed grains microstructure was obtained. Equiaxed grains of the SLM processed GTD222 may be achieved by using a higher solution treatment temperature and longer time. The investigation of higher solution heat treatment temperature and time on the microstructure of the SLM processed GTD222 will be performed soon to further optimize the heat treatment process of the alloy.

The SEM images showing the microstructure of the etched as-built and HTed GTD222 samples were shown in Figure 3. The microstructure of the as-built GTD222 was a typical cell and columnar structure, which was similar to the AM processed superalloys. No γ’ or carbide was precipitated in the cell structures due to the high cooling rate, as shown in Figure 3a. In the HTed GTD222, the cell structures were disappeared and many carbides were precipitated in the γ, as highlighted in Figure 3b. The γ’ was also observed in the sample, as indicated in Figure 3b. 

The substructure of the as-built GTD222 was shown in Figure 4. Qualitatively, the dislocation density in the as-built GTD222 was high and dislocations were tangled at the boundaries of the columnar, as shown in Figure 4a. The method determines the dislocation density was introduced in the literature [26]. The selected area electron diffraction of the green circle area marked in Figure 4a shows that no precipitate was formed in the as-built GTD222.

In the HTed GTD222, the tangled dislocations were disappeared and γ’ was precipitated, as shown in Figure 5a,b. The precipitated γ’ was further identified by the dark-field image and the selected area electron diffraction of green circle area marked in Figure 5b, as displayed in Figure 5c,d.

In addition, the EDS results shown in Figure 6 confirmed that carbides were also precipitated along the boundaries of the columnar. Moreover, most of the carbides were smaller than 200 nm, much smaller than carbides precipitated in the casted GTD222 [27].

Typical tensile strain-stress curves of the HTed GTD222 samples were showed in Figure 7.

Tensile properties of the as-built GTD222 were also tested for comparison. The yield strength (YS), ultimate tensile strength (UTS) and elongation have been summarized and listed in Table 1.

The UTS and YS of the HTed GTD222 at room temperature are 1424 ± 7 MPa and 1120 ± 6 MPa, respectively. The UTS and YS of the as-built GTD222 at ambient temperature were 1100 ± 7 MPa and 831 ± 5 MPa, respectively. However, the elongation of the HTed GTD222 was inferior to as-built GTD222, which is decreased from 24.3 ± 2.5% to 13.7 ± 1.4%. The fracture morphologies of the as-built and HTed GTD222 samples were displayed in Figure 8.

The fracture of as-built GTD222 was consisting of uniformly distributed dimples and tear ridges. Most of the dimples were bigger than 1 μm. Similarly, dimples and tear ridges on the fracture of the HTed GTD222 were homogeneous. However, the dimples on the HTed GTD222 fracture were smaller and shallower. The smaller dimples suggesting that the deformation ability of the HTed GTD222 matrix was inferior [28,29]. Declined ductility of HTed GTD222 may be caused by the presence of brittle carbides in γ matrix, which acts as crack nucleation sites under tensile stress. The fracture morphologies were consistent with the inferior elongation of the HTed GTD222. Dimples and tear ridges indicated that the main fracture mechanism of the as-built GTD222 was ductility fracture. The fracture mechanism of the HTed GTD222 was similar to that of the as-built GTD222. mentioned above, the strengths of the HTed GTD222 were higher than that of its as-built GTD222 counterparts. The difference in strengths was due to the microstructure evolution of the HTed GTD222 samples. As can be seeing from Figure 3, γ’ and nano-scaled carbides were precipitated in the HTed GTD222, which will impede the dislocations moving resulting in a higher dislocation density during the deformation process [30]. As consequence, the HTed samples have higher YS than their as-built counterparts, although the initial dislocation density of the HTed GTD222 was lower than its as-built counterparts.

## 4. Conclusions

Microstructure evolution of the SLM processed GTD222 and mechanical properties of as-built and HTed samples were investigated in the current work. The main conclusions were summarized as follows:The microstructure of the HTed GTD222 were consisting of columnar grains, 1170 °C solution + 800 °C aging 1170 °C.A large amount of γ’ and nano-scaled carbides were precipitated in the HTed GTD222. The γ’ precipitates were homogeneously distributed in the matrix while the carbides were distributed along the boundaries of the columnar.The high yield strength of the HTed GTD222 was 1120 ± 6 MPa, which was caused by the precipitation of the γ’ and the nano-scaled carbides.

## Figures and Tables

**Figure 1 materials-14-03668-f001:**
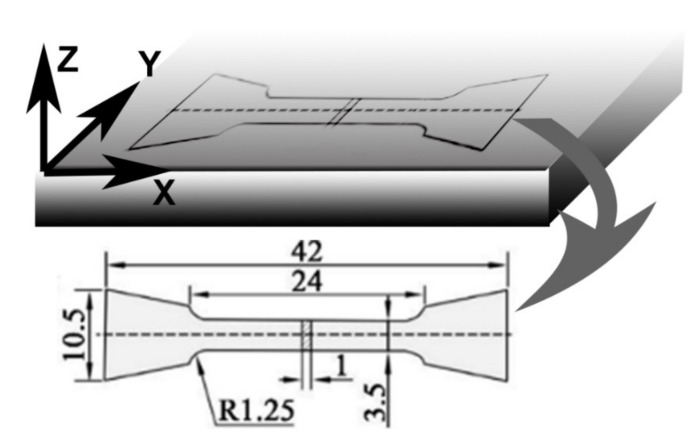
The dimension (in mm) of the tensile samples, Z was the building direction.

**Figure 2 materials-14-03668-f002:**
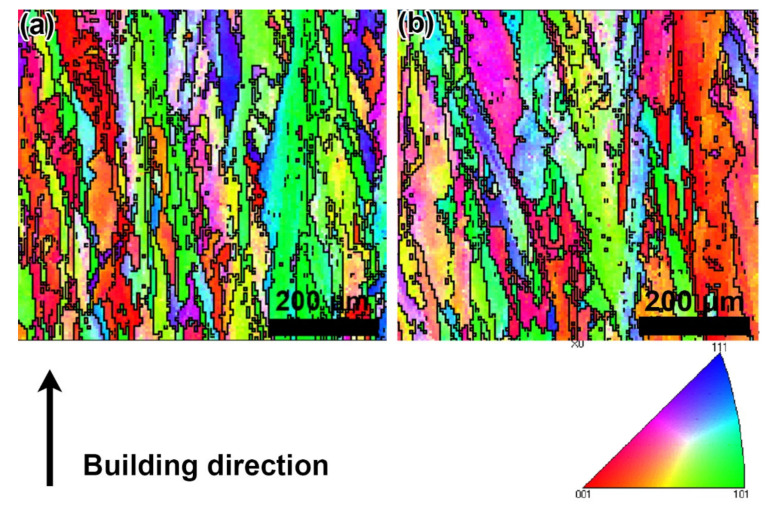
The IPF images of the GTD222. (**a**) as-built sample, (**b**) HTed sample.

**Figure 3 materials-14-03668-f003:**
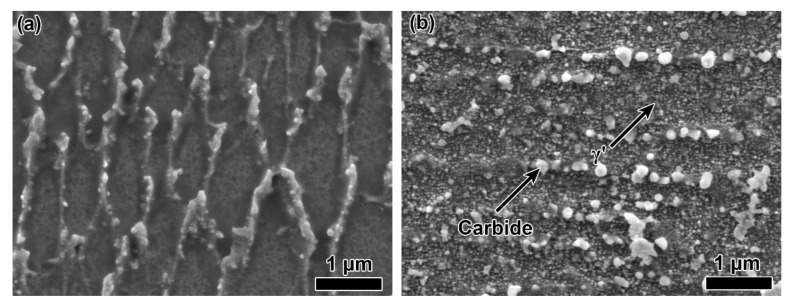
SEM images showing microstructure of the GTD222 superalloy: (**a**) as-built sample, (**b**) HTed sample.

**Figure 4 materials-14-03668-f004:**
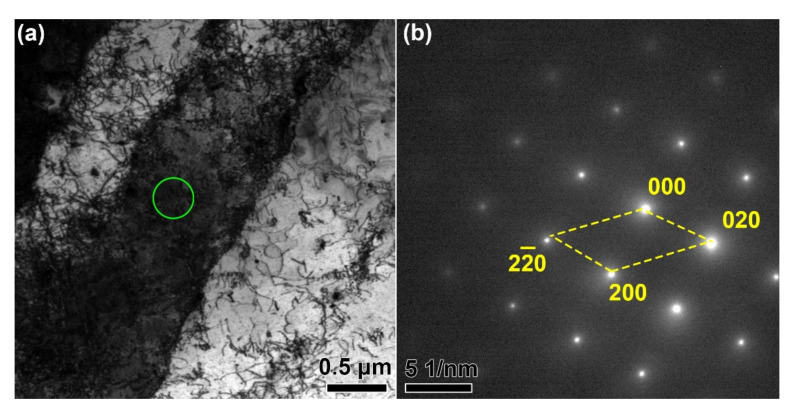
TEM images showing the microstructure of as-built GTD222. (**a**) The bright field image of the GTD222, (**b**) selected area electron diffraction of green circle area marked in (**a**).

**Figure 5 materials-14-03668-f005:**
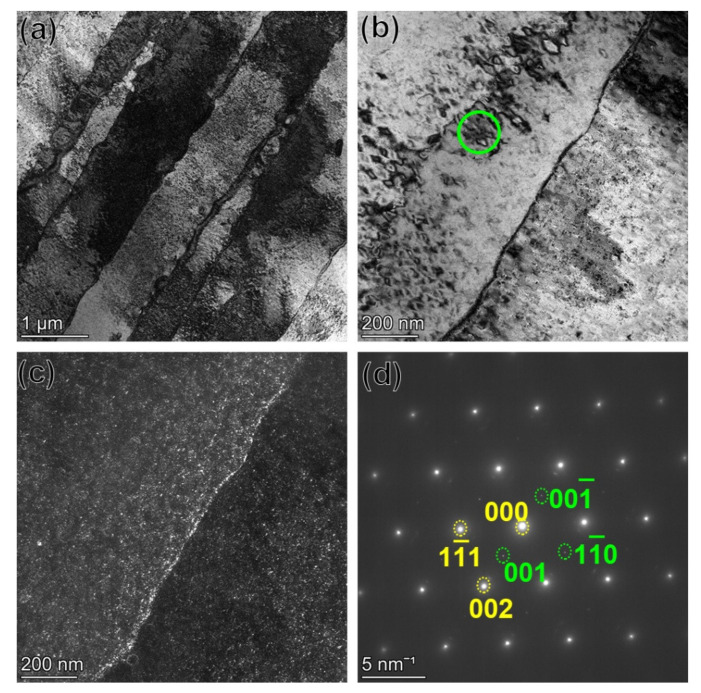
The microstructure of the SLM GTD222 after standard heat-treated. (**a**) Columnar structure, (**b**) enlarged columnar structure, (**c**) dark-field image of the columnar structure, (**d**) selected area electron diffraction of green circle area marked in (**b**), the γ’ and γ were indexed using green and yellow font, respectively.

**Figure 6 materials-14-03668-f006:**
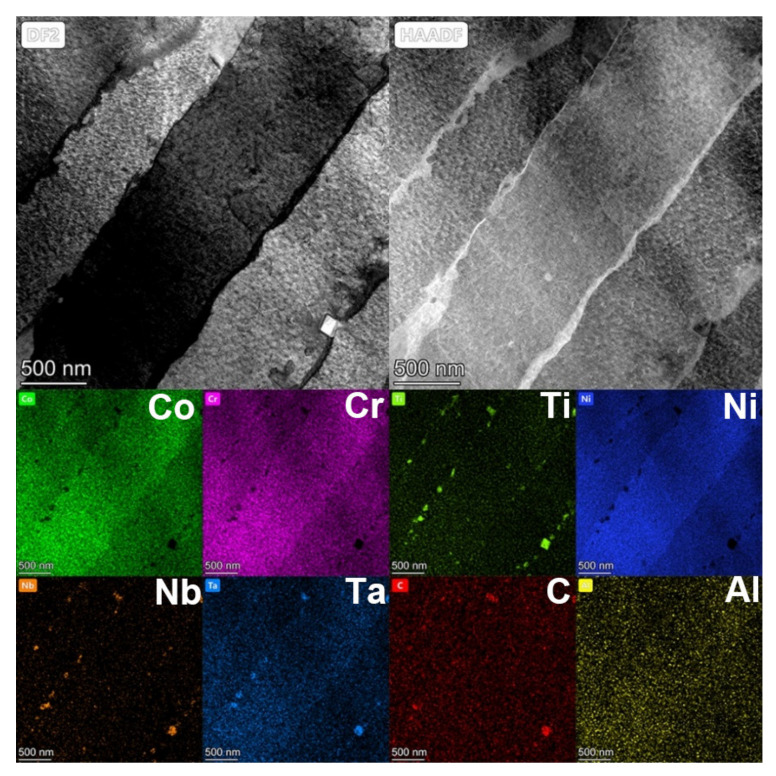
TEM and EDS images of the SLM processed GTD222 after heat treatment. Dark Field (DF), High-angle Annular Dark Field (HADDF).

**Figure 7 materials-14-03668-f007:**
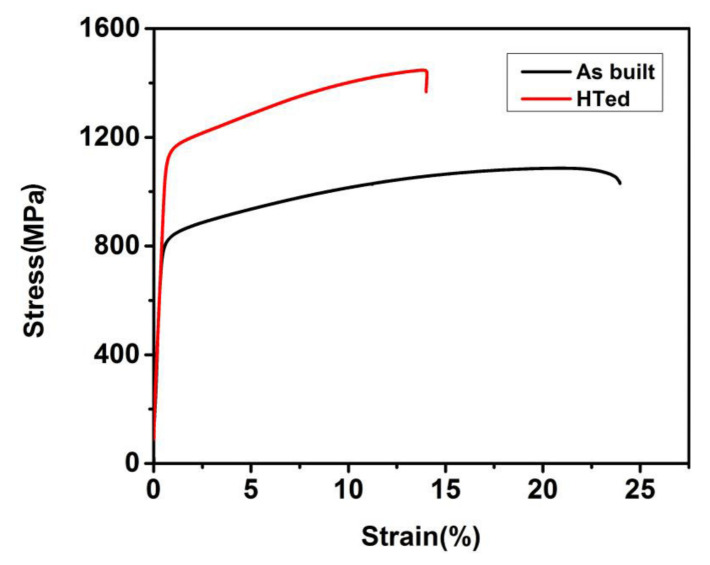
Typical tensile strain-stress curves of the GTD222 at different states.

**Figure 8 materials-14-03668-f008:**
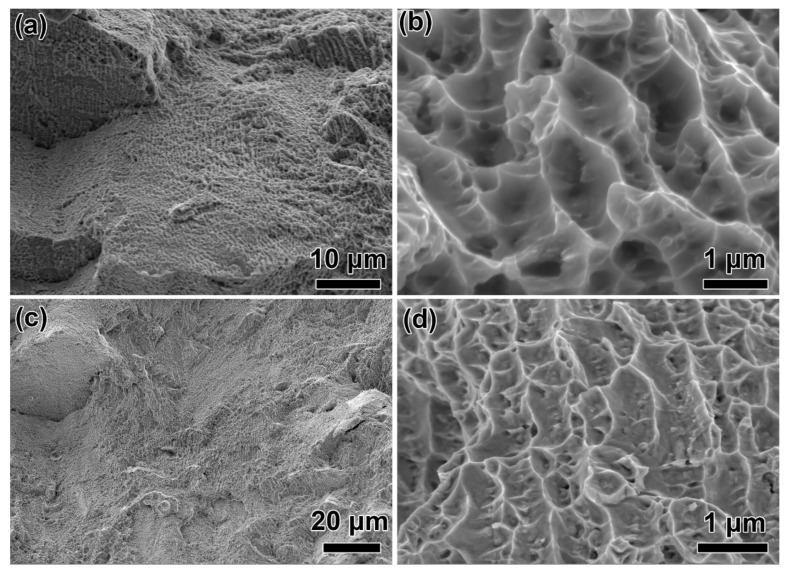
Fracture of the as-built and HTed GTD222. (**a**,**b**) as-built, (**c**,**d**) HTed GTD222.

**Table 1 materials-14-03668-t001:** The mechanical properties of the as-built and HTed GTD222. Yield strength (YS), ultimate tensile strength (UTS).

Materials	YS (MPa)	UTS (MPa)	Elongation (%)
As-built GTD222	831 ± 5	1100 ± 7	24.3 ± 2.5
HTed GTD222	1120 ± 6	1424 ± 7	13.7 ± 1.4

## Data Availability

All the data is available within the manuscript.

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
