# Peer review of "Effect of Heat Treatment on Microstructure and Mechanical Properties of a Selective Laser Melting Processed Ni-Based Superalloy GTD222"

_materials, 2021, doi:10.3390/ma14133668_

Round 1

Reviewer 1 Report

The study investigates the effect of heat treatment on microstructure and mechanical properties of SLM processed Ni-based superalloys GTD222. The paper, which is very compact, covers an important issue and is overall well written. The study is worth doing and the results are useful – especially the characterization of the material conditions. However, there is a very limited amount of results or rather material conditions: one material condition in as-built and one in heat treated - with only one heat-treated parameter set (solution and aging).

1) It is necessary to explain, why these heat treatment parameter set is used?

2) Why did you not investigate the solution heat-treated condition, too? …would be very helpful.

3) It is mentioned that higher solution heat treatment temperature and time will (may!) cause equiaxed grains – why did you not check up on this!!! …here more valuable data would exist for discussion.

I would highly recommend to perform some more tests with additional heat treatment parameter settings.

Please find some more comments and suggestions:

- abstract: please remove the last sentence, you only investigated one heat treatment condition… guidance you can call it when increasing time or temperature causes a change in microstructure and mech. properties and you suggest rather increasing or decreasing time or temperature, you concentrated only on one HTed condition, only

- do not use abbreviations in the abstract: AM and HTed

- start with abbreviations in the main text: SLM, HTed – however, AC you are using only twice, here is no need on short version AC

- does the aging time of 8 hours cause peak-hardening? …what happens and shorter or longer aging time?

- the sentence “Some researchers believe that… “ need revision. By just using one reference it is only one study… and they do not believe, they will have found out that…

- please mention why equiaxed grains were/are helpful to uniform mech. properties – this would give a better understanding

- how did you determine the dislocation density? …most likely just by appearance in the TEM images: so it would be needed to mention this. …furthermore: the dislocation density was high compared to … you need a reference for high, low and normal

- please call the curves: stress-strain curves, it would be advised to present the mech. properties YS, TS and Elongation in a Table

- please check the sentence: The difference in strengths was stem from… stem?

- the conclusion is far too short and should only based on you results and not assumption from the literature (equiaxed grains may be…), here you need to add some information on your microstructure according to your heat treatment, mention fracture behavior and mech. properties

Author Response

Reviewer #1: The study investigates the effect of heat treatment on microstructure and mechanical properties of SLM processed Ni-based superalloys GTD222. The paper, which is very compact, covers an important issue and is overall well written. The study is worth doing and the results are useful – especially the characterization of the material conditions. However, there is a very limited amount of results or rather material conditions: one material condition in as-built and one in heat treated - with only one heat-treated parameter set (solution and aging).

1) It is necessary to explain, why these heat treatment parameter set is used?

Response: GTD222 was a mature Ni-based superalloy and heat treatment process of the casting GTD222 has been optimized. The heat treatment parameters used in the current study were the optimized heat treatment parameters suitable for the casting GTD222. It’s rational to speculate that these heat treatment parameters will adjust the microstructure of the additive manufactured GTD222 into a reasonable condition.

2) Why did you not investigate the solution heat-treated condition, too? …would be very helpful.

Response: Yes, thanks for your suggestion. The solution heat-treated can alleviate the element segregation in the as-built Ni-based superalloys and is important to the properties of the alloy. The investigation of the solution heat-treated influence on the microstructure of GTD222 will be performed in the further work soon.

3) It is mentioned that higher solution heat treatment temperature and time will (may!) cause equiaxed grains – why did you not check up on this!!! …here more valuable data would exist for discussion. I would highly recommend to perform some more tests with additional heat treatment parameter settings.

Response: Thanks for your recommendation. The equiaxed grains of the AM processed GTD222 may achieved through solution heat treatment or hot isostatic pressing progresses and this work is what we are performing. We will report those results in the future.

Please find some more comments and suggestions:

- abstract: please remove the last sentence, you only investigated one heat treatment condition… guidance you can call it when increasing time or temperature causes a change in microstructure and mech. properties and you suggest rather increasing or decreasing time or temperature, you concentrated only on one HTed condition, only

Response: Thanks for your suggestion, the last sentence in the Abstract has been removed.

- do not use abbreviations in the abstract: AM and HTed

Response: Yes, the abbreviations in the abstract have been revised.

- start with abbreviations in the main text: SLM, HTed – however, AC you are using only twice, here is no need on short version AC

Response: Yes, the 'AC' has been rewritten as 'air cooling'.

- does the aging time of 8 hours cause peak-hardening? …what happens and shorter or longer aging time?

Response: In general, the aging time has great influence on the strengths of the Ni-based superalloys for the size and volume of strengthening precipitates γ'. In our previous work, the influence of aging temperature (800-950 oC), direct aging without solution, on the mechanical properties of the AM processed GTD222 has been discussed. The yield strength and ultimate tensile strength of the GTD222 fabricate with SLM increased with decreasing ageing temperature, while the elongation declined simultaneously. So, it is reasonable to speculate that the solution treatment +aging treated AM processed GTD222 may displaying same microstructure evolution trend with direct aging AM processed GTD222.

- the sentence “Some researchers believe that… “ need revision. By just using one reference it is only one study… and they do not believe, they will have found out that…

Response: the sentence has been revised in the Manuscript.

- please mention why equiaxed grains were/are helpful to uniform mech. properties – this would give a better understanding

Response: Yes, the feature of equiaxed grains has been illustrated in the Results and discussion section paragraph 1 line 18 as follows: ''eliminating anisotropy of alloy and make its properties more predictable''.

- how did you determine the dislocation density? …most likely just by appearance in the TEM images: so it would be needed to mention this. …furthermore: the dislocation density was high compared to … you need a reference for high, low and normal

Response: Yes, the dislocation density of the alloys was qualitative analysis and has been illustrated in the Manuscript.

- please call the curves: stress-strain curves, it would be advised to present the mech. properties YS, TS and Elongation in a Table

Response: Yes, the curves have been revised as the stress-strain curves, the YS, UTS and Elongation have been listed in Table 1 as follows.

- please check the sentence: The difference in strengths was stem from… stem?

Response: Yes, the sentence has been rewritten as '' The difference in strengths was due to… ''.

- the conclusion is far too short and should only based on you results and not assumption from the literature (equiaxed grains may be…), here you need to add some information on your microstructure according to your heat treatment, mention fracture behavior and mech. Properties.

Response: Yes, the Conclusions has been revised.

Reviewer 2 Report

The article presents a topic about the Effect of heat treatment on microstructure and mechanical properties of a selective laser melting, however there are points that need to be clarified and summarized.

  1. Abstract: I suggest improving the abstract, making clear the purpose and the main results that have been achieved with this study.
  2. Introduction: The introduction is very general and very short. It should be improved and clearly explain the purpose and final objective of this study with appropriate references. In a research paper, it is expected that introduction section briefly explains the starting background and, even more important, the originality (novelty) and relevancy of the study is well established. Once this is done, hypothesis and objectives of the study need to be addressed, as well as a brief justification of the conducted methodology.
  3. Experimental: Explain in greater detail the experimental processes carried out in the research.
  4. Results and discussions: Put in this section only the part of the results obtained in this research for in another separate section (discussion), compare and discuss those results explaining the main advances obtained by comparing and discussing them with the results obtained by the studies of other authors..
  5. Discussion Section: Create a separate Discussion section. The Discussion section should compare the study by clearly comparing the results obtained by the authors with other studies conducted by other authors.
  6. Conclusions Section: Improve the conclusions section, it is very general and does not clearly explain the main objectives achieved in this research. The conclusions section should present in a clear and summarized way the main parts obtained with this study and the main contributions.

Author Response

Reviewer #2: The article presents a topic about the Effect of heat treatment on microstructure and mechanical properties of a selective laser melting, however there are points that need to be clarified and summarized.

Abstract: I suggest improving the abstract, making clear the purpose and the main results that have been achieved with this study.

Response: Yes, the Abstract has been revised. the purpose and the main results that have been clarified as follows: ''This study provides essential microstructure and mechanical properties information for optimizing the heat treatment process of the AM processed GTD222. ''.

Introduction: The introduction is very general and very short. It should be improved and clearly explain the purpose and final objective of this study with appropriate references. In a research paper, it is expected that introduction section briefly explains the starting background and, even more important, the originality (novelty) and relevancy of the study is well established. Once this is done, hypothesis and objectives of the study need to be addressed, as well as a brief justification of the conducted methodology.

Response: Yes, the Introduction has been extended. The background and originality are summarized as follows: ''Ramakrishnan et al. found that the mechanical properties of the AM processed Haynes 282 superalloy were superior to its casting counterparts. The increased volume of car-bide is regarded as the most influential factors. In our previous work, the effect of direct aging on the microstructure and mechanical properties of the AM processed GTD222 was carried out. The results show that the strengths of the GTD222 increased with decreasing ageing temperature, while the elongation declined simultaneously. The precipitation of γ' was affected by the aging but no carbides were precipitated after direct aging. The effect of the carbides on mechanical properties was not clear yet. '' and ''This work can aid in optimizing the heat treatment process of the SLM processed GTD222. ''.

Experimental: Explain in greater detail the experimental processes carried out in the research.

Response: Yes, more details have been added in the Experimental section.

Results and discussions: Put in this section only the part of the results obtained in this research for in another separate section (discussion), compare and discuss those results explaining the main advances obtained by comparing and discussing them with the results obtained by the studies of other authors.

Discussion Section: Create a separate Discussion section. The Discussion section should compare the study by clearly comparing the results obtained by the authors with other studies conducted by other authors.

Response: Thanks for your suggestion. Writing separately allows us to focus on presenting the results. However, sometimes the results and the discussion are inherently related. Hence, we choose stating the results while analyzing and discussing them.

Conclusions Section: Improve the conclusions section, it is very general and does not clearly explain the main objectives achieved in this research. The conclusions section should present in a clear and summarized way the main parts obtained with this study and the main contributions.

Response: Yes, the Conclusions have been revised.

Reviewer 3 Report

Dear authors,

You did an interesting work but the presentation of it does not satisfy. My remarks are as follows:

The introduction section does not provide sufficient background analysis and relevant references are missing. Namely, there should be more analysis of articles about heat treatment and microstructures-mechanical properties of Ni-based super alloys.

How did you determine the conditions for heat treatment?

Why did you choose 760 °C for tensile test?

Please indicate how did you perform the etching of the samples for SEM?

The description of the metallographic procedure for microstructural analysis (SEM/EBSD, TEM) should be more explain.

You said that at least five samples were tensile tested and you gave an averages values. But you did not emphasize did you analyse a microstructure of all samples and was it the same/similar.

Conclusion should be supplemented with more results of mechanical test.

Please explain/emphasize a value of this research and its importance.

Nowhere is porosity mentioned, and it is inevitable in SLM.

Please add explanations into the manuscript.

Author Response

Reviewer #3: You did an interesting work but the presentation of it does not satisfy. My remarks are as follows:

The introduction section does not provide sufficient background analysis and relevant references are missing. Namely, there should be more analysis of articles about heat treatment and microstructures-mechanical properties of Ni-based super alloys.

How did you determine the conditions for heat treatment?

Response: GTD222 was a mature Ni-based superalloy and heat treatment process of the casting GTD222 has been optimized. The heat treatment parameters used in the current study were the optimized heat treatment parameters suitable for the casting GTD222.

Why did you choose 760 °C for tensile test?

Response: The GTD222 is a mature casting Ni-based superalloy and can retain excellent mechanical properties at high temperatures up to 760 ℃. Hence, we choose 760 °C for tensile test in order to comparing with casting GTD222.

Please indicate how did you perform the etching of the samples for SEM?

Response: The SEM samples were etched in a Cr2O3 corrosive liquid at 7 V for 3 s. The EBSD samples were polished using vibration polishing machine with diamond polishing fluid for 2 hours. The information has been added in the Experimental section.

The description of the metallographic procedure for microstructural analysis (SEM/EBSD, TEM) should be more explain.

Response: Yes, the missing information were added in the Experimental section.

You said that at least five samples were tensile tested and you gave an averages values. But you did not emphasize did you analyse a microstructure of all samples and was it the same/similar.

Response: Yes, thanks for your reminding. In the current study, all the tensile samples have similar microstructure status and this condition has illustrated in the Manuscript.

Conclusion should be supplemented with more results of mechanical test.

Response: Yes, the Conclusions have been rewritten to make it more logical.

Please explain/emphasize a value of this research and its importance.

Response: Yes, this work can aid in optimizing the heat treatment process of the SLM processed GTD222. The value of this research has been emphasized in the Introduction section.

Nowhere is porosity mentioned, and it is inevitable in SLM. Please add explanations into the manuscript.

Response: thanks for your reminding. Porosity of the AM processed GTD222 studied in the current work has been discussed in the Manuscript.

Reviewer 4 Report

The authors studied the properties of GTD222 fabricated by selective laser melting. While the manuscript is generally well executed, there are several issues in the manuscript that should be addressed before further consideration for publication. 1. Suggest to use ISO/ASTM terminology to describe the process. There should be a more in depth description of the process as well, as the significant of the process on the results is not highlighted - Sing et al. (2021), Emerging Metallic Systems for Additive Manufacturing: In-situ Alloying and Multi-metal Processing in Laser Powder Bed Fusion, Progress in Materials Science 119, 100795 - Tian et al. (2020), A Review on Laser Powder Bed Fusion of Inconel 625 Nickel-Based Alloy, Applied Sciences 10 (1), 81 - Sing & Yeong (2020), Laser powder bed fusion for metal additive manufacturing: perspectives on recent developments, Virtual and Physical Prototyping 15 (3), 359-370 2. There is anisotropy on the properties of the parts produced by SLM. Any consideration and discussion on this? 3. How many replicates were used in the study? Were the samples fabricated homogenous? SLM is known to produce microstructure variation even across a single sample. 4. Any discussion on the difference between the as build and heat treated samples? Are the changes significant?

Author Response

Reviewer #4: The authors studied the properties of GTD222 fabricated by selective laser melting. While the manuscript is generally well executed, there are several issues in the manuscript that should be addressed before further consideration for publication.

  1. Suggest to use ISO/ASTM terminology to describe the process. There should be a more in depth description of the process as well, as the significant of the process on the results is not highlighted - Sing et al. (2021), Emerging Metallic Systems for Additive Manufacturing: In-situ Alloying and Multi-metal Processing in Laser Powder Bed Fusion, Progress in Materials Science 119, 100795 - Tian et al. (2020), A Review on Laser Powder Bed Fusion of Inconel 625 Nickel-Based Alloy, Applied Sciences 10 (1), 81 - Sing & Yeong (2020), Laser powder bed fusion for metal additive manufacturing: perspectives on recent developments, Virtual and Physical Prototyping 15 (3), 359-370

Response: Yes, the description of the process has been revised.

  1. There is anisotropy on the properties of the parts produced by SLM. Any consideration and discussion on this?

Response: Yes, anisotropy is common in the SLM processed superalloys. In general, the mechanical properties of the specimens built with their cylinder axis (loading direction) oriented either parallel to the building direction are superior to specimens built perpendicular to the building direction. In the current study, all the specimens are perpendicular to the building direction. By doing this, the mechanical properties of the specimens built with their cylinder axis (loading direction) oriented either parallel to the building direction can be speculated rationally.

  1. How many replicates were used in the study? Were the samples fabricated homogenous? SLM is known to produce microstructure variation even across a single sample.

Response: At least 5 specimens are tested for each type of alloy. According to the tensile results, the difference in mechanical properties between different tensile samples is tiny. It is reasonable to predict that the microstructures of the GTD222 at different states are homogeneous at the macro level, even though nonuniform microstructure may existing in some areas.

  1. Any discussion on the difference between the as build and heat treated samples? Are the changes significant?

Response: Yes, the microstructure and the mechanical properties difference between as build and heat-treated samples are discussed in the Manuscript. On one side, the dislocation density of the as build samples is higher than the heat-treated samples. On the other side, the γ' and nano-scaled carbides are precipitated in the heat-treated samples. The γ' and nano-scaled carbides will impede the dislocations moving resulting in higher strengths of the heat-treated samples.

Round 2

Reviewer 1 Report

Thank you for addressing comments and suggestion.

- it is important, that you explain in the paper, why heat treatment parameter set is used – it would be of interest of the reader, not only for me

- I would highly recommend to mention, that further work on higher solution heat treatment temperature and time will be reported soon – otherwise the reader wonders as much as I did, why this has not done in this study!

- I will remain at my question: how did you determine the dislocation density? According to your response qualitative analysis has been done and illustrated in the manuscript – please explain in the paper!

- conclusion: 3. The high strengths of the HTed GTD222 at …. MPa and …. MPa were caused…. (values would be good!)

Author Response

Responses to the referees’ comments

We thank the reviewers for their efforts and time in reviewing the manuscript and making insightful comments and suggestions which are valuable for improving the quality and readability of our manuscript. We thank the comments and suggestions very much, and have taken the following actions to address them:

Reviewers' comments:

Reviewer #1: Thank you for addressing comments and suggestion.

- it is important, that you explain in the paper, why heat treatment parameter set is used – it would be of interest of the reader, not only for me

Response: Yes, the reason why heat treatment parameter set is used has been illustrated in the Experimental Section paragraph 3 as follows: '' The heat treatment parameters used in the current study were the optimized heat treatment parameters suitable for the casting GTD222.''.

- I would highly recommend to mention, that further work on higher solution heat treatment temperature and time will be reported soon – otherwise the reader wonders as much as I did, why this has not done in this study!

Response: Yes, the plan to investigate the higher solution heat treatment temperature and time on the microstructure of the SLM processed GTD222 was illustrated in the Manuscript as follows: '' The investigation of higher solution heat treatment temperature and time on the microstructure of the SLM processed GTD222 will be performed soon to further optimize the heat treatment process of the alloy.''.

- I will remain at my question: how did you determine the dislocation density? According to your response qualitative analysis has been done and illustrated in the manuscript – please explain in the paper!

Response: Yes, the method was illustrated and related paper was cited in the Manuscript as follows: '' The method determines the dislocation density was introduced in the literature [26].''.

- conclusion: 3. The high strengths of the HTed GTD222 at …. MPa and …. MPa were caused…. (values would be good!)

Response: Yes, thanks for your suggestion. The conclusion 3 has been rewritten as follows: ''The high yield strengths of the HTed GTD222 can be attributed to was 1120±6 MPa, which was caused by the precipitation of the γ' and the nano-scaled carbides. ''.

Reviewer 3 Report

The manuscript has been greatly improved. Thank you for accepting my comments. Best regards

Author Response

Thanks for your comments and suggestions very much.

Reviewer 4 Report

NIL

Author Response

(The authors gave the same response as above.)
